# Asymmetric electric field screening in van der Waals heterostructures

Lu Hua Li [1], Tian Tian [2], Qiran Cai[1], Chih-Jen Shih [2] & Elton J.G. Santos [3]

A long-standing challenge facing the combination of two-dimensional crystals into hetero-junction is the unknown effect of mixing layer of different electronic properties (semi-conductors, metals, insulators) on the screening features of the fabricated device platforms including their functionality. Here we use a compelling set of theoretical and experimental techniques to elucidate the intrinsic dielectric screening properties of heterostructures formed by $MoS_2$ and graphene layers. We experimentally observed an asymmetric field screening effect relative to the polarization of the applied gate bias into the surface. Sur-prisingly, such behavior allows selection of the electronic states that screen external fields, and it can be enhanced with increasing of the number of layers of the semiconducting $MoS_2$ rather than the semi-metal. This work not only provides unique insights on the screening properties of a vast amount of heterojunction fabricated so far, but also uncovers the great potential of controlling a fundamental property for device applications.

[1] Institute for Frontier Materials, Deakin University, Geelong Waurn Ponds Campus, Victoria 3216, Australia. [2] Institute for Chemical and Bioengineering, ETH Zürich, 8093 Zürich, Switzerland. [3] School of Mathematics and Physics, Queen's University Belfast, Belfast BT7 1NN, United Kingdom. Correspondence and requests for materials should be addressed to E.J.G.S. (email: e.santos@qub.ac.uk)

The van der Waals heterostructures (vdWHs) composed of two-dimensional (2D) crystals and precisely assembled in a deterministic order constitutes a remarkable paradigm for promising electronic and optoelectronic applications with enhanced features and performance[1–6]. With the properties of the individual layers being characterized since the discovery of graphene and other 2D materials[7], the great challenge is how to combine them in order to obtain unusual physical and chemical phenomena not observed on the original sheets[1]. The continuous development of experimental methods that allow atomically thin materials to be fabricated on-demand and to be placed on desired locations with an unprecedented control and accuracy have opened pioneering avenues to fabricate complex device architectures using regular bottom-up approaches[2–4]. The atomic flatness and lack of dangling bonds at the surface of 2D layered materials, such as graphene, boron nitride (BN) and transition metal dichalcogenides (TMDCs), allow them to form non-covalent interactions with a wide range of materials without the condition of lattice matching that normally heterojunctions would have. One of the first realizations of such vdWHs was the fabrication of BN/graphene interfaces and the probing of their electronic properties using gate bias[8]. Such system remarkably showed that graphene could develop superior electrical properties, achieving levels of performance comparable to those observed with freestanding layers. In this assembly, BN layers work as an underlying substrate screening out graphene from any dangling bonds, corrugations and charge inhomogeneities that are inherent to standards SiO$_2$ surfaces to play a role on its electronic properties[8,9]. A direct extension of this system was the complete encapsulation of graphene, TMDCs and the combination of them between BN layers[5,6,10,11]. This rapidly showed that such device framework displayed improved transport properties with high-mobilities[10]. It is important to remark that in all these systems electrostatic gating schemes have been one of the main driving force to probe the chemical and physical properties of either isolated 2D materials before assembling, or their vdWHs afterwards. Electric gate bias measurements have become a feasible way to control, induce and manipulate electronic properties of almost any vdW crystals since graphene early days[12–14]. This relatively simple setup allows deep insight into the charge density reorganization between different stacking sequences, and on the electric-field screening, which determines most of the exquisite device properties observed in vdWHs. However, no direct measurements of the electrostatic screening features of different combinations of 2D materials have been reported yet, which ramp up further understanding and developments of vdWHs into any technological platform.

Here we show that the dielectric properties of an archetypal vdWH, involving graphene and MoS$_2$, displays an asymmetrical behavior relative to the polarization of the applied gate bias into the surface allowing an asymmetric screening. Using electric force microscopy (EFM), we observe that graphene/MoS$_2$ heterostructures exhibit distinct electric response to external fields with respect to the stacking sequence and the direction of electric field. A large depolarization field is recorded at the MoS$_2$ side, which is shown to be dependent majority on the number of MoS$_2$ layers into the heterostructure. A multiscale theoretical framework is developed for elucidating such stacking-dependent asymmetric electric screening phenomenon in graphene/MoS$_2$ vdWHs. At the atomistic level, we employed quantum mechanical ab initio simulations based on density functional theory (DFT) using a total energy vdW functional to resolve various electronic properties of the vdWHs, including polarization, charge transfer and band structures. Our results indicate that the asymmetric formation of interfacial dipole moments preferably at the MoS$_2$ facet accounts for the asymmetric response. We further show that such asymmetric behavior can be explained using a classical quantum capacitor model, described by a set of self-consistent electrostatic

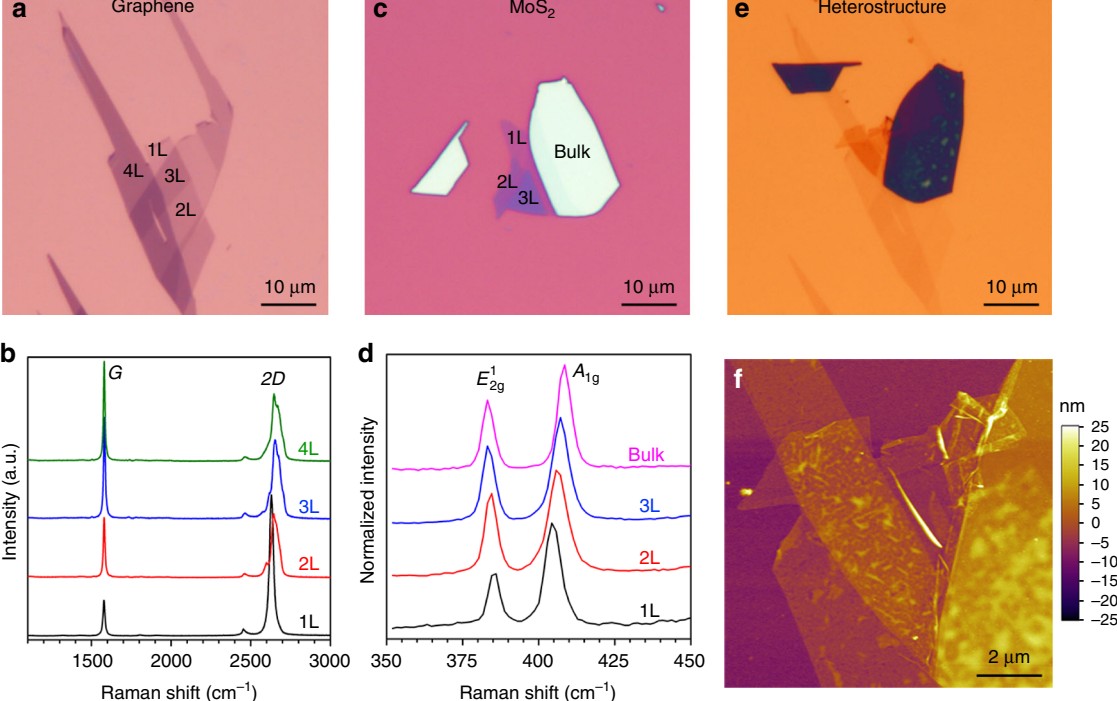

**Fig. 1** Sample fabrication and optical characterization. **a** Optical microscopy photo of the mechanically exfoliated graphene of different thicknesses on 90 nm SiO$_2$/Si; (**b**) the corresponding Raman spectra (514.5 nm wavelength); (**c**) optical microscopy photo of the as-exfoliated MoS$_2$ on 270 nm SiO$_2$/Si; (**d**) the corresponding Raman spectra; (**e**) optical microscopy photo of the MoS$_2$ (top)/graphene (bottom) heterostructures transferred onto 100 nm-thick Au coated SiO$_2$/Si by PMMA method; (**f**) the corresponding AFM image of the heterostructure

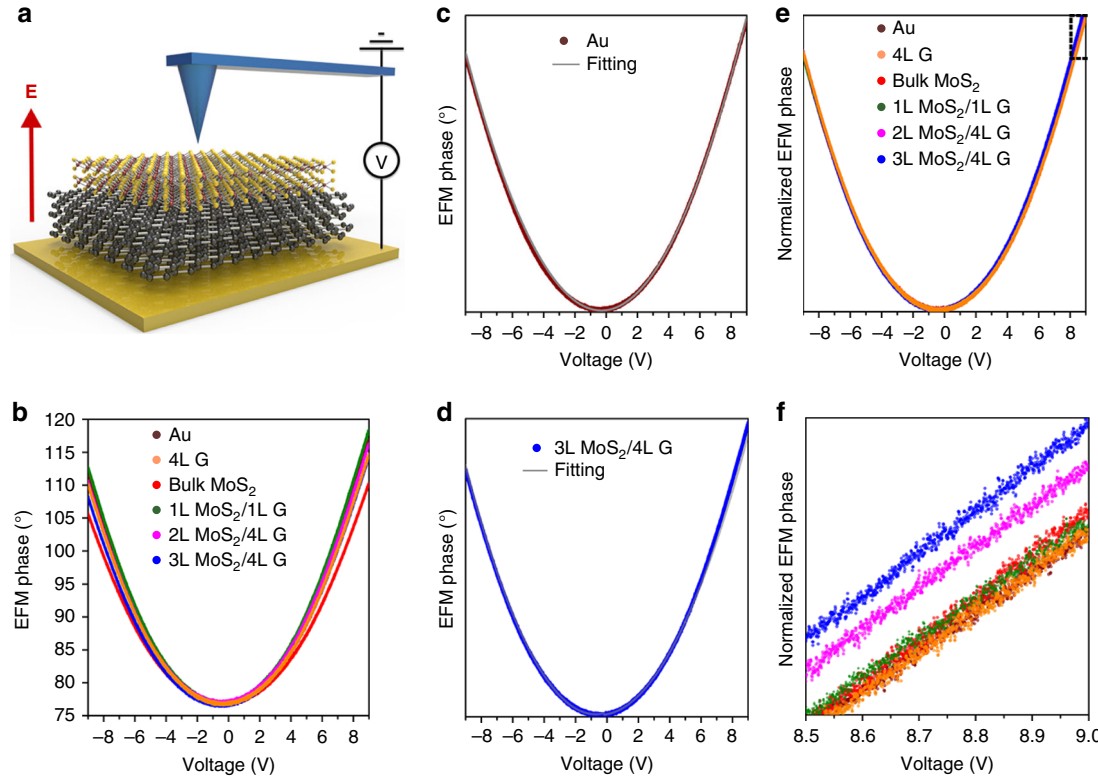

**Fig. 2** EFM response of vdW heterostructures under different electric bias. **a** Drawing of the EFM spectroscopy setup with the Au substrate underneath graphene and MoS₂ layers. Positive (negative) bias generates electric fields towards MoS₂ (graphene), which are detected by the cantilever. **b** The raw EFM data from the six samples under a sweeping substrate DC voltage (+9 to −9 V); **c** The excellent fitting of the EFM phase spectrum from the Au substrate (brown) using second-degree polynomials (gray); **d** The asymmetry in the EFM phase spectrum from the 3L MoS₂/4L Graphene heterostructure demonstrated using the same fitting process; **e** Normalized EFM phase spectra from the six samples to show the asymmetric parabolas obtained from some of the heterostructures; **f** Enlarged view of the dashed area in (**e**)

conservation equations and treating the 2D layers according to their individual electronic genomes (i.e., energy levels, band gap and quantum capacitance). Our theoretical framework has good consistency with the experimental results at both ab initio and classical levels, showing that the combinations of 2D materials with distinct electronic structures can result in vdWHs with rich screening features. Furthermore, our theoretical framework is readily applicable for other vdWHs beyond graphene/MoS₂ to explore a wide range of 2D material combinations with programmable electronic screening properties, which may greatly benefit the design of 2D vdWH-based functional devices.

## Results

**Characterization and measurements.** Heterostructures of graphene and MoS₂ of different thicknesses on Au coated Si wafer were achieved by two rounds of polymethyl-methacrylate (PMMA) transfer[15]. Figure 1a shows the optical image of 1–4L thick graphene nanosheets mechanically exfoliated on a SiO₂ (90 nm)/Si substrate. Their thickness was confirmed by the Raman spectra (514.5 nm wavelength) as shown in Fig. 1b. The intensity of the 2D band of the 1L graphene was much stronger than that of the G-band; while the intensity of the 2D and G bands of the 2L graphene is comparable. The Raman results are consistent with previous reports[16]. The absence of Raman D band suggests the high quality of the graphene. Atomically thin MoS₂ nanosheets were exfoliated on a SiO₂ (270 nm)/Si substrate (Fig. 1c). The portions with different purplish optical contrast gave $E_{2g}^1$ and $A_{1g}$ Raman bands centered at 385.3 and 404.7 cm⁻¹, 384.3 and 406.3 cm⁻¹, and 383.6 and 407.2 cm⁻¹, respectively (Fig. 1d). Therefore, they corresponded to 1–3L MoS₂[17,18]. To fabricate MoS₂/

Graphene heterostructures, the MoS₂ was first transferred onto the graphene with the help of PMMA, and then the MoS₂/Graphene structure was relocated onto a 100 nm-thick Au coated SiO₂/Si substrate, as shown by the optical microscopy photo in Fig. 1e. The corresponding atomic force microscopy (AFM) image of the heterostructure on Au is displayed in Fig. 1f.

We used EFM to measure the electric field screening properties of the MoS₂/Graphene heterostructures. EFM has been used to investigate the electric field screening in graphene, BN, and MoS₂ nanosheets[19–21]. However, the experimental setup in the current study was slightly different from those in these previous reports. We applied a DC voltage sweeping from +9 to −9 V to the conductive cantilever, while the Au substrate was grounded, which generated external electric fields of different intensities. The tip of the conductive cantilever was oscillating at its first resonant frequency stayed at a few nanometers above the heterostructures, and acted as a sensor monitoring the capacitance change caused by the change of electric susceptibility of the heterostructures (Fig. 2a). Subtle capacitance changes could be detected by EFM phase shift (Δϕ), which can be described as[21,22]:

$$\Delta\phi = \frac{\partial F/\partial z}{k} \cdot Q_{cant},  \quad (1)$$

where $\partial F/\partial z$ is the local force gradient, representing the derivative of the electrostatic force felt by the cantilever tip; $k$ is the spring constant of the cantilever; and $Q_{cant}$ is the Q factor of the cantilever. For simplicity, the interaction between the cantilever tip and the sample in EFM is often viewed as an ideal capacitance (see Supplementary Note 1 and Supplementary Figure 1). Therefore, the local force gradient due to the capacitive

interaction becomes[21,22]:

$$\partial F/\partial z = \frac{1}{2}\frac{\partial^2 C}{\partial z^2}(V + V_{CPD})^2, \qquad (2)$$

where $C$ and $z$ are the local capacitance and distance between the tip and the sample, respectively; $V$ is the tip voltage; $V_{CPD}$ is the contact potential difference (CPD) due to the mismatch of the work functions between the tip and the sample. The raw data of the EFM spectroscopy from the Au substrate, 4L graphene, bulk $MoS_2$, and three heterostructures, namely 1L $MoS_2$/1L Graphene, 2L $MoS_2$/4L Graphene, and 3L $MoS_2$/4L Graphene are shown in Fig. 2b. The EFM phase of all the samples formed opening-up parabolas with the axis symmetry parallel with the $y$-axis as a function of the DC voltage. Considering the quadratic function in Eq. 2, we were not surprised that the parabolas were recorded. As $V_{CPD}$ was fixed for each sample, the formation of the parabolic curves was due to the sweeping $V$. In other words, $V$ was a dominant parameter in our EFM measurements. The opening-up means attractive capacitive interactions under both positive and negative voltages[21]. However, the different samples gave rise to slightly different shapes of the parabolas. This was caused by the other parameters, especially $C$ and $z$. The distance $z$ was inevitably slightly different from sample to sample during the EFM spectroscopy measurements. Although the local capacitance $C$ is very difficult to define, as it depends on many factors, including the shape, size, conductivity, and dielectric property of a sample and cantilever, $C$ was different from sample to sample. Therefore, it is understandable that the different samples resulted in the slightly different parabolic shapes in EFM spectroscopy.

Intriguingly, we found that the parabolas of the EFM phase from some heterostructures lacked the mirror-symmetry as that of a perfect parabola. We give typical examples on symmetric and

asymmetric EFM phase in Fig. 2c, d, respectively. The EFM data from the Au substrate could be well fitted by the second-degree polynomial (gray vs. brown in Fig. 2c), but the same fitting process was not able to reproduce the right part (positive bias) of the EFM phase curve from 3L $MoS_2$/4L Graphene (gray vs. blue in Fig. 2d). To compare this phenomenon from the different samples, we normalized all the EFM data, as shown in Fig. 2e. The enlarged view from the dashed area in Fig. 2e is displayed in Fig. 2f. Similar to the Au substrate, the 4L graphene, bulk $MoS_2$, and 1L $MoS_2$/1L Graphene gave rise to symmetric parabolas of the EFM phase; in contrast, the heterostructures of few-layer $MoS_2$ and graphene, i.e., 2L $MoS_2$/4L Graphene and 3L $MoS_2$/4L Graphene, showed deviated-parabolic EFM phase values under more positive voltages. These EFM results were peculiar. As discussed previously, most of the parameters determining EFM phase, except $V$, should be constant during each measurement. The asymmetric parabolic EFM curves suggest that the electric field screening properties of 2L $MoS_2$/4L Graphene and 3L $MoS_2$/4L Graphene were probably not constant under different voltages, i.e. external electric fields. In turn, the local capacitance $C$ should change slightly accordingly. This phenomenon was not shown in Au film, 4L graphene, bulk $MoS_2$, or 1L $MoS_2$/1L Graphene, but became prominent in the heterostructures with increased thickness of graphene and $MoS_2$. We tried to qualitatively analyze the EFM results to estimate the change of electric susceptibility ($\chi$) of the 3L $MoS_2$/4L Graphene heterostructure (Supplementary Figure 2). The behavior displays an increment of $\chi$ with the bias pointing to the $MoS_2$ surface. In the following sections, we will show that such asymmetric behavior can be explained at two theoretical levels, both by ab initio simulations with vdW-functionals, as well as a quantum-capacitance-based classical electrostatic model.

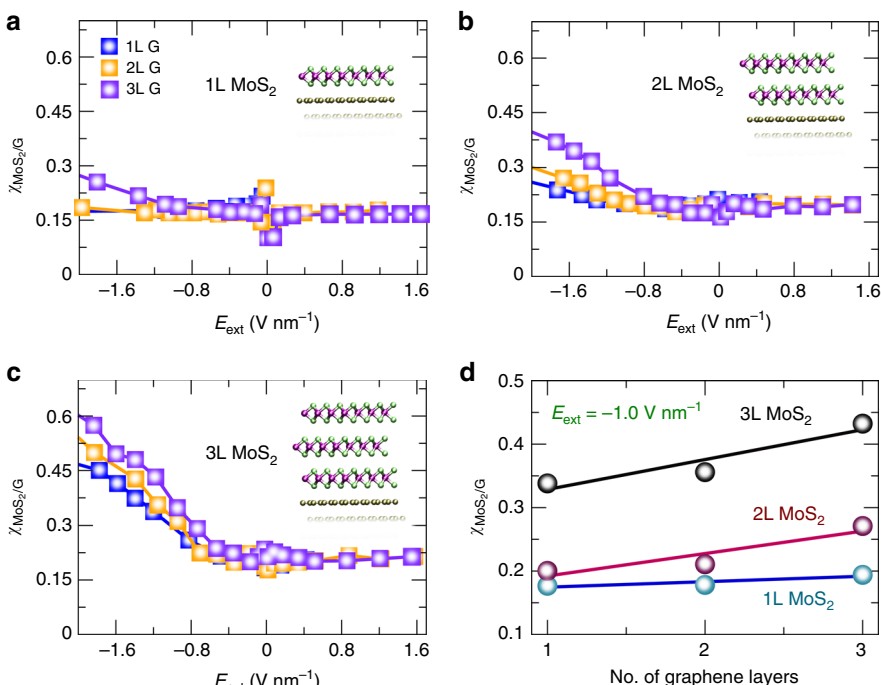

**Fig. 3** Ab initio vdW first-principles calculations for $MoS_2$/Graphene heterostructures. **a–c** Electric susceptibility $\chi_{G/MoS_2}$ as a function of external electric fields $E_{ext}$ (V nm$^{-1}$) at 1L, 2L and 3L $MoS_2$, respectively (see insets). The number of graphene layers systematically increases in each panel at a fixed number of $MoS_2$ sheets following the labeling shown in (**a**). The polarization of the field follows the orientation in Fig. 2a, where positive (negative) fields go towards graphene ($MoS_2$) firstly. **d** $\chi_{G/MoS_2}$ as a function of the number of graphene layers at a static field of $-1.0$ V nm$^{-1}$. Different curves correspond to different number of $MoS_2$ layers on the each vdW-heterostructures. Straight lines are fitting curves using a linear equation ($\chi = A_0 + A_1 N$), where the angular coefficients are: 1L (0.008), 2L (0.035) and 3L (0.047)

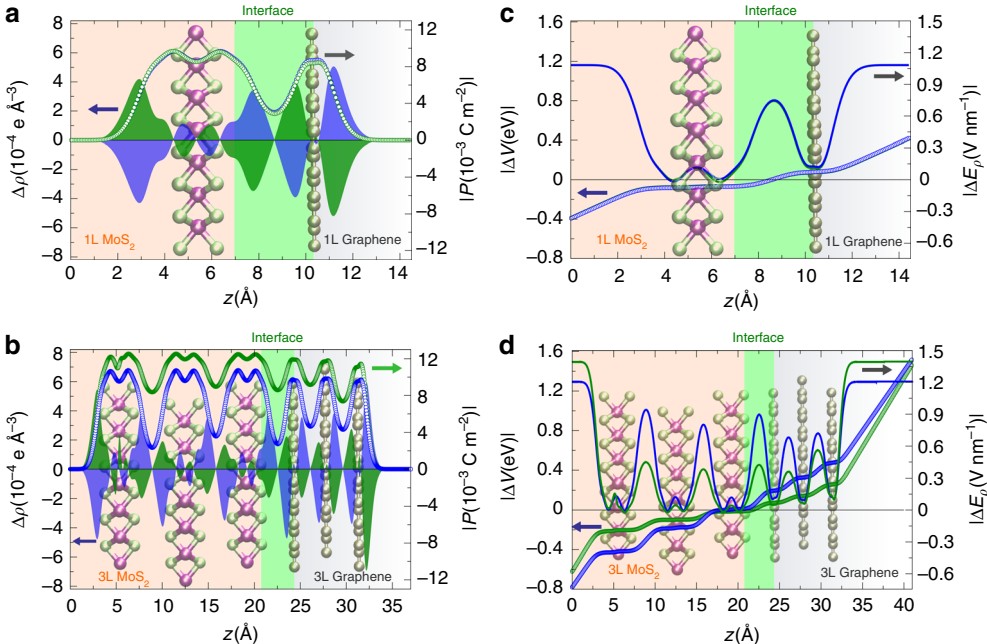

**Fig. 4** Asymmetric dipolar contributions at the MoS$_2$/Graphene interface. **a**, **b** Induced charge density $\Delta\rho\left(10^{-4}\,\text{eÅ}^{-3}\right)$ (left y-axis) and electric polarization $\left|P\left(10^{-3}\,\text{Cm}^{-2}\right)\right|$ (right y-axis) for 1L MoS$_2$/1L Graphene and 3L MoS$_2$/3 L Graphene, respectively. The applied electric field is ±1.0 V nm$^{-1}$. Blue (green) curves correspond to positive (negative) fields. Positive (negative) fields go towards graphene (MoS$_2$), and vice versa. The MoS$_2$/Graphene interface is highlighted to show the unbalanced formation of electric dipole moments between graphene and MoS$_2$ accordingly with the number of layer layers used to form the heterostructures. (**c-d**) Difference in electrostatic potential $\Delta V(\text{eV}) = V(E_{\text{ext}} \neq 0) - V(E_{\text{ext}} = 0)$ in the slabs with and without the external electric field of ±1.0 V nm$^{-1}$, and their corresponding response field $\Delta E_\rho\left(V\,\text{nm}^{-1}\right)$ for 1L MoS$_2$/1L Graphene and 3L MoS$_2$/3L Graphene, respectively. The absolute values of $|\Delta E_\rho|$ and $|\Delta V|$ are taking in (**c**) and (**d**) for comparison at the same side of the plot. Geometries for all systems are highlighted at the background of each panel in opacity tone

**Quantum mechanical first-principle simulations.** To better understand this intriguing phenomenon, we performed two levels of theretical analysis using quantum mechanical ab initio calculations based on density functional theory (DFT); and a classical electrostatic approach using a capacitance model based on charge conservation equation solved variationally (see Methods for details). We first address the quantum mechanical part of the MoS$_2$/Graphene heterostructures. We calculated the degree of polarization in the vdW heterostructures of graphene and MoS$_2$ of different number of layers in response to the applied electric fields in terms of the electric susceptibility $\chi_{\text{G/MoS}_2}$. We used the quantum mechanical model presented in refs.[23,24], where a fully based ab initio approach was employed to extract information about the dielectric response at finite-electric fields and large supercells. No external parameters apart from the magnitude of the external electric fields were utilized in a self-consistent calculation. The simulations also took into account vdW dispersion forces, electrostatic interactions, and exchange-correlation potential within DFT at the same footing.

Figure 3a–c show the variation of $\chi_{\text{G/MoS}_2}$ with the electric field at 1L, 2L and 3L MoS$_2$, respectively, but with a distinct number of graphene layers. Strikingly, only negative bias affected $\chi_{\text{G/MoS}_2}$ despite of the number of graphene and MoS$_2$ sheets present in the heterostructures. This is in remarkable agreement with the experimental results, where an asymmetrical response was recorded only from graphene and few-layer MoS$_2$ heterostructures (Fig. 2f and Supplementary Figure 2). The effect is enhanced, as more MoS$_2$ layers are included into each graphene system. The largest increment is noticed on 3 L MoS$_2$/3 L Graphene, where a four-fold enhanced magnitude of $\chi_{\text{G/MoS}_2}$ relative to zero field was calculated (Fig. 3c). We also observed that graphene layers have minor contributions to the effect, as $\chi_{\text{G/MoS}_2}$ slightly varied as more graphene was putted together on

top of MoS$_2$ (Fig. 3d). At a fixed value of the electric field ($-1.0$ V/nm), larger asymmetric screening was displayed as the number of MoS$_2$ layers increased: the screening in the heterostructures containing 3L MoS$_2$ was almost doubled that of 1L and 2L MoS$_2$ heterostructures. The slope of $\chi_{\text{G/MoS}_2}$ vs. the number of graphene layers also increased with the thickness of MoS$_2$ (1L (0.008), 2L (0.03) and 3L (0.05)), which indicates that thicker MoS$_2$ tends to be more correlated with variations in the number of graphene layers. This follows the behavior observed from EFM measurements, which heterostructures involving thicker MoS$_2$ sheets in contact with graphene gave rise to a more asymmetric EFM phase parabola (Fig. 2f). On the basis of these results, it becomes clear that the transition metal dichalcogenide layers play a key role on this screening effect. We will analyze in the following the modifications of the electronic structure of the heterostructures at finite electric fields, and elucidate the origin of this asymmetric susceptibility dependence on the external bias.

Figure 4 shows that the behavior of $\chi_{\text{G/MoS}_2}$ with the electric bias results from the asymmetrical polarization $P$ associated to which side of the heterostructure the field interacts first. At negative bias, a larger amount of induced charge $\Delta\rho$ was displaced towards the surface-layers of the MoS$_2$ in the heterostructure, which consequently generates a polarization $P$ that provided a better screening to the external electric fields relative to positive bias (Fig. 4a, b). As the number of MoS$_2$ layer is small, little differences are noticed under the reversed electric field, as the dipole moment formed at the interface roughly compensated each other (Fig. 4a). This effect is enlarged, as thicker MoS$_2$ sheets are included. This was due to the amount of interfacial charge redistribution, which generated electric dipole moments preferentially aligned along one direction (Fig. 4b). Electric fields pointing towards graphene were not well screened as those towards MoS$_2$ layers because the induced polarization was not so

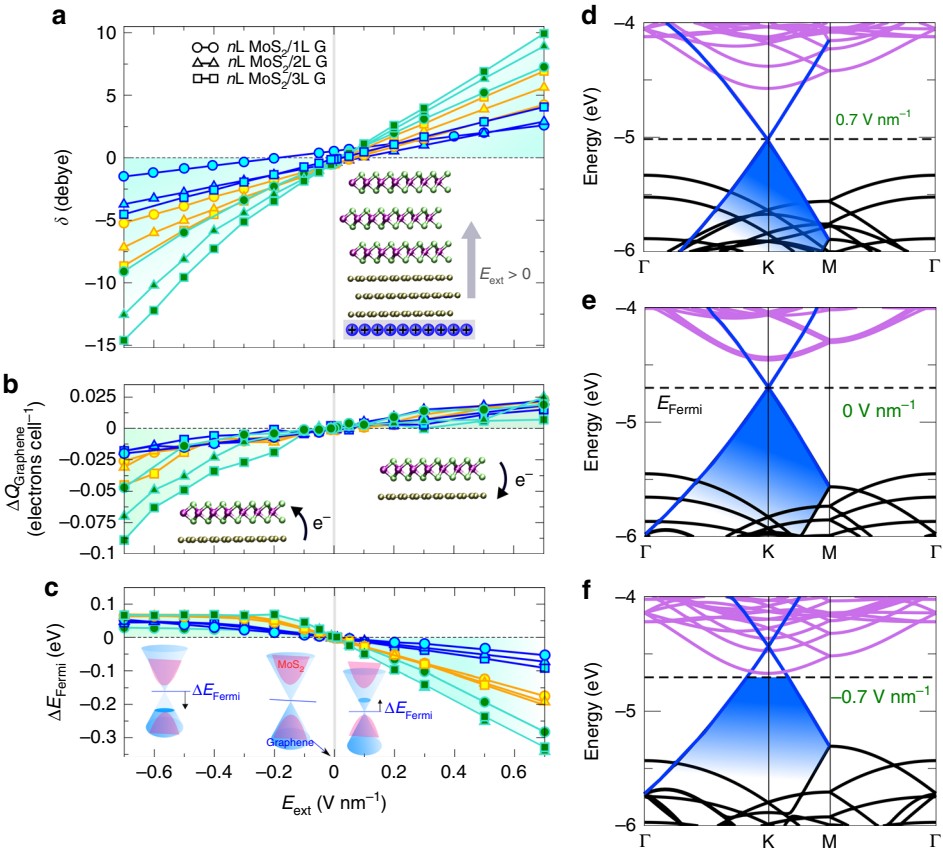

**Fig. 5** Asymmetric electronic response under external electric fields. **a–c** Electric dipole moment $\delta$ (Debye), charge transfer $\Delta Q_{Graphene}$(electrons per cell), between graphene and $MoS_2$, and Fermi level variation, $\Delta E_{Fermi}$(eV), as a function of electric field $E_{ext}$(V nm$^{-1}$), respectively. The different curves and colors correspond to the different number of graphene and $MoS_2$ layers. Blue (1L $MoS_2$), orange (2L $MoS_2$) and green (3L $MoS_2$). The same labeling in (**a**) for the number of graphene layers and colors for the $MoS_2$ are used throughout the different plots. Positive (negative) fields point towards graphene ($MoS_2$) layers as shown in the inset in (**a**). The calculated charge transfer $\Delta Q_{Graphene}$ follows the trend displayed in the insets in (**b**). That is, positive bias induces charge transfer from $MoS_2$ to graphene, and vice versa. The insets in (**c**) summarize the effect of the electric field on $\Delta E_{Fermi}$(eV) and the resulting electronic structure: positive (negative) bias shifts downward (upward) in energy the Fermi level. Also notice the relative shifts of graphene and $MoS_2$ states with the electric bias. **d–f** Electronic band structures of the 3L $MoS_2$/1L Graphene heterostructure at different gate bias. Graphene states are highlighted in blue and $MoS_2$ bands in faint pink. Fermi level is shown by the dashed-line in each panel. An asymmetrical dependence of the electronic properties with the electric field is noted in all calculated quantities

efficient to generate response fields $\Delta E_\rho$ that would shield the heterostructures completely. This means that higher magnitudes of electric field were observed inside thinner heterostructures, rather than thicker ones (Fig. 4c, d). We also observed that the induced electric potential $|\Delta V|$ shows a smooth variation over the interface, and it is almost independent of the number of layers composing the junction. $|\Delta V|$ displays high magnitudes over fields towards graphene layers, with a change in polarity at the $MoS_2$ layer near the interface with $|\Delta V| = 0$ for thicker vdWHs (see Fig. 4d). This indicates that the interfacial-charge balance in both systems that generates $|\Delta V|$ is sensitive to the amount of polarization charge from the $MoS_2$ layers. That is, the thicker the $MoS_2$ sheets, the larger the polarization. A consequence of this electric field direction-dependent polarization is noted in the different magnitudes of $\Delta E_\rho$ observed in the vacuum region outside of 3L $MoS_2$/3L Graphene system for positive and negative fields (Fig. 4d). In electrostatic boundary conditions, where the normal component of the displacement field $D$ has to be preserved into the system[25], it gives:

$$D = E_{vacuum} = E_{slab} + 4\pi P_{slab},\qquad(3)$$

where $E_{slab}$ corresponds to the field in the sheets and $P_{slab}$ to the induced polarization. It is worth noting that $P_{slab}$ differs to $P$

because the latter is calculated directly from the average induced charge using the Poisson equation and the former directly from the boundary conditions and the input field in the simulations (see "Methods" section for details). For negative fields towards $MoS_2$ layers, the second term on the right-hand side in Eq. 3 involving the polarization is appreciably large, which generates a depolarization or response field $\Delta E_\rho$ that would overcome the applied external bias. This resulted in smaller electric fields inside the heterojunction (Fig. 4d). A similar effect is observed to fields directed to graphene layers, but higher in magnitudes inside the sheet due to smaller induced polarization. The polarization at the $MoS_2$/Graphene heterostructure is therefore a contributing factor in the special screening field effect we measured by EFM.

On the basis of the previous analysis, several implications on the electronic structure of the heterojunction can be foreseen. Figure 5 shows an asymmetric electronic response with the external electric field for several quantities. At negative bias, the induced dipole moments associated to the S–Mo–S bonds displace the charge towards the surface of the $MoS_2$ sheet (Fig. 5a), which generated charge transfer from graphene to $MoS_2$ at the interface (Fig. 5b). This charge rearrangement is smaller for positive fields because of the semiconducting nature of the $MoS_2$ layer with less charge-carriers on its surface, and the semi-

metallic character of graphene. This results in less polarizable field-dependent facet, smaller charge-transfer from $MoS_2$ to graphene, and consequently better screening. The effect of the electric field can also be noted on the tuning of the Fermi level $\Delta E_{Fermi}$ relative to the charge-neutrality point when no doping

concentration was considered on either graphene or $MoS_2$ (Fig. 5c). Thin systems (e.g., 1L $MoS_2$/1L Graphene) tend to tune their Fermi level almost linearly with the electric field, which is a property intrinsically present at the pristine layers[26–28]. As the number of $MoS_2$ sheets increased, $\Delta E_{Fermi}$ displayed

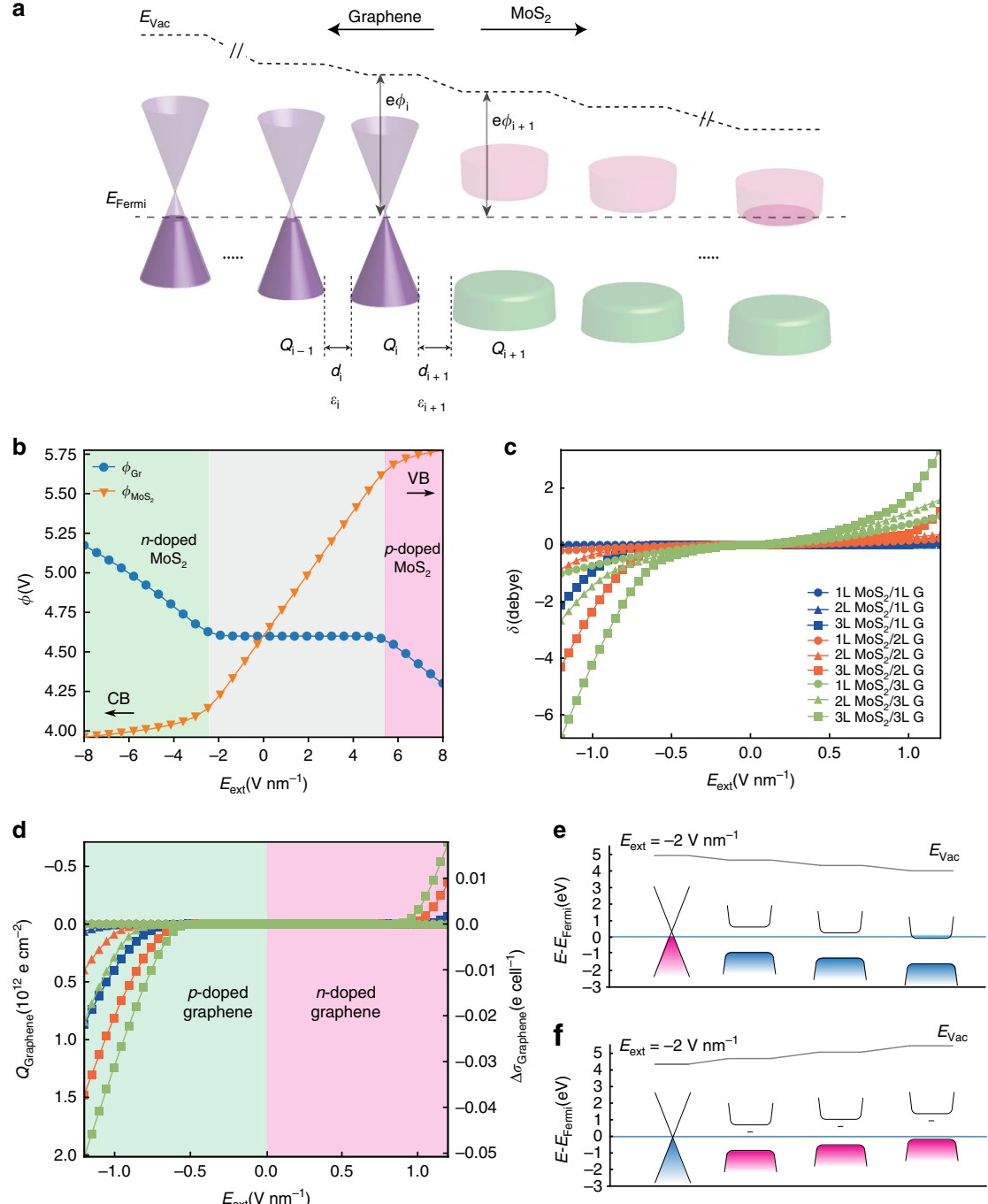

**Fig. 6** Classical electrostatic approach for charge redistribution of $MoS_2$/Graphene vdW heterostructures. **a** Schematic band diagram of the multilayer $MoS_2$/Graphene heterostructures. Vacuum level ($E_{vac}$), work function ($\phi_i$), surface charge density ($Q_i$), interlayer distance ($d_i$), and relative permittivity ($\varepsilon_i$) are shown for each stacking $i$. **b** Work functions of graphene and $MoS_2$ as functions of external electric field $E_{ext}$ (V nm$^{-1}$). The regimes of the n-doped and p-doped $MoS_2$ are highlighted in green and faint red, respectively. The Fermi level reaches the conduction band (CB) or valence band (VB) of $MoS_2$ when large negative or positive $E_{ext}$ is applied, relatively (as shown by the arrows). **c** Electric dipole moment $\delta$ (Debye) as a function of $E_{ext}$ (V nm$^{-1}$) for different number of graphene and $MoS_2$ layers on the heterostructures. **d** Surface charge density $Q_{Graphene}$ (left axis) and the amount of charge transfer between graphene and $MoS_2$ layers $\Delta\sigma_{Graphene}$ (right axis) as a function of $E_{ext}$ (V nm$^{-1}$) for different $MoS_2$/Graphene systems. The curves follow the labeling in (**c**). **e–f** Band alignments for an illustrative case, e.g., 3L $MoS_2$ /1L Graphene under −2.0 V nm$^{-1}$ and +2.0 V nm$^{-1}$ electric fields, respectively. Positive charges are shown in faint red and negative charges are shown in blue, respectively. Only band structures near the Fermi level are shown for illustration

variations at positive fields as large as 0.34 eV for 3L MoS$_2$/3L Graphene but almost negligible for the negative electric field with the formation of a plateau at $-0.2$ V nm$^{-1}$ and beyond. Such asymmetric behavior has been observed when MoS$_2$ layers are used in metal-insulator-semiconductor junctions[29]. Carrier doping induced by the electric field was responsible for the variation of the Fermi level or the work function of MoS$_2$, mainly along one direction, which is directly related to the unbalance of charge density between both sides of the semi-metal and the semiconductor interface. This indicates that the intrinsic character of the electronic structure of each system in vdW heterostructures contributes to the formation of the asymmetric screening observed. This effect has several main implications on the fundamental electronic structure of the MoS$_2$/Graphene interfaces as can be appreciated on the band structure calculated at different magnitudes of gate bias for a sample system (e.g. 3 L MoS$_2$/1L Graphene) in Fig. 5d–f. Similar trends are observed for different thicknesses of graphene and MoS$_2$. At 0.0 V nm$^{-1}$, the Fermi level crossed the Dirac point of graphene, as no charge-imbalance was present between both systems. Several MoS$_2$ states at the conduction band were observed at 247.86 meV relative to the Fermi level, which also corresponds to the Schottky barrier presents at the interface[30]. At finite fields, those states are observed to shift up (down) with positive (negative) electric bias, with a consequent split as large as ~0.20 eV. This modifies their occupation, as some graphene states can become occupied (positive bias) or unoccupied (negative bias) according to the field polarization. The insets in Fig. 5c summarize the main effect of the bias on graphene and MoS$_2$ states near the Fermi level. This indicates that for electric fields toward the dichalcogenide layer, the states mainly composed of the conduction band of MoS$_2$ with minor contribution from graphene were responsible for the charge-screening effect and vice versa. This suggests the important role of the interface on the electrical properties of the vdW heterostructures, as the polarity of the electric field can select, which states can screen the system against external bias.

**Classical electrostatic approach**. Apart from the mighty ab initio approach that gives the full picture of electronic states in the vdWHs, it is more desirable that such asymmetric behavior of the vdWHs can be modeled inexpensively using several key electronic properties from the individual 2D material layers. Here we show that the asymmetric electronic screening of MoS$_2$/Graphene vdW heterostructures under an external electric field can be well described using a classical electrostatic model, taking the quantum capacitance into consideration[31]. As the 2D vdWHs are stacked via non-covalent interactions, it is found that the individual properties of 2D materials can still be largely preserved in their stacked layers, which are coupled by the Coulombic interactions[32]. The idea behind our classical electrostatic model is that each individual layer has its own electronic "genome" (i.e., energy level, band gap, quantum capacitance) extracted from ab initio calculations, which can be used as building blocks in solving the electrostatic conservation equations of the whole vdWH, under the non-coupling assumption. Figure 6a schematically shows the band diagram of the MoS$_2$/Graphene vdWH used in the classical model. Due to the fact that there is no electron drift in the vdWH at equilibrium (which is consistent with the EFM experimental setup and ab initio configurations), the Fermi level $E_{Fermi}$ aligns throughout the MoS$_2$/Graphene vdWH. For simplicity of the model, we further assume that (i) the density of states (DOS) of individual layer is invariable with the stacking order and the external electric field and (ii) the interlayer distances ($d_i$) are not affected by the external electric field. Note that although the transition of band structure is ignored in assumption (i), it has

been shown that such classical treatment using Coulombic coupling has relatively high consistency with the ab initio simulations[32]. The charge and potential distribution in the vdWH is solved by several conservation equations in a self-consistent approach[33]: (i) the charge of individual 2D layer $Q_i$ follows the charge conservation of the vdWH, (ii) $Q_i$ is determined by the electric displacement field adjacent to the 2D layer, (iii) $Q_i$ determines the work function of layer $i$, $\phi_i$ and (iv) the work function difference between two adjacent layers is determined by the electric displacement field. More details about the mean-field model can be found in Methods. For the simplest case of 1L MoS$_2$/ 1L Graphene, we plot in Fig. 6b the work functions $\phi_i$ of both materials, as a function of the external electric field strength ($E_{ext}$) ranging from $-8$ to $+8$ V nm$^{-1}$. Highly n-doped and p-doped MoS$_2$ regimes can be found when $E_{ext} < -2.3$ V nm$^{-1}$ or $E_{ext} > 5.5$ V nm$^{-1}$, respectively. The fermi level of MoS$_2$ shifts close to its conduction band (CB) or valence band (VB) in both regimes, respectively, which is accounted for the charge accumulation in the vdWH. Note that a noticeable charge accumulation ($>10^{12}$ e cm$^{-2}$) occurs even when the Fermi level of MoS$_2$ is still ca 0.1 eV away from the band edges, due to the fact that the low quantum capacitance of graphene near its intrinsic Fermi level. On the other hand, when $E_{ext}$ is between $-2.3$ and 5.5 V nm$^{-1}$, the Fermi level of MoS$_2$ lies far away from the band edges, resulting that the vdWH is merely not polarized and the work function of graphene $\phi_{Gr}$ shows little change with $E_{ext}$. We find that the polarization of the MoS$_2$/Graphene vdWH is more enhanced under negative external electric field, which corresponds with the findings in Fig. 3a. We ascribe such asymmetry to the difference between the electronic structures of graphene and MoS$_2$: MoS$_2$ is considered as a *n*-type semiconductor with its intrinsic Fermi level ($-4.5$ eV) closer to its CB ($-4.0$ eV) than its VB ($-5.8$ eV)[34–36], while graphene has symmetric linear band structure around its Dirac point ($-4.6$ eV)[37], all energy levels are compared with the vacuum level that is set at 0 eV. Due to their close Fermi level values, little charge transfer occurs between graphene and MoS$_2$ under weak electric field, and the degree of charge transfer is mainly determined by the position of the Fermi level with respect to the CB or VB of MoS$_2$. Following the same procedure, we calculated the dipole moment $\delta$ (Fig. 6c), and the charge density of the graphene layers $Q_{Graphene}$ (Fig. 6d) for a different number of layers at the MoS$_2$/Graphene interface. The results from the classical model show good consistency compared with the quantum mechanical *ab initio* calculations of dipole moment (Fig. 5a) and charge transfer (Fig. 5b): the charge redistribution is more pronounced with increased layer numbers of graphene and MoS$_2$, and the MoS$_2$ layers contributes more to such effect than graphene. Note that for thicker graphene layers (e.g., nL MoS$_2$/3L Graphene), a considerable amount of total dipole moment can still be observed under weak electric field ($|E_{ext}| < 0.5$ V nm$^{-1}$), when the charge transfer between graphene and MoS$_2$ is negligible ($<10^{11}$ e cm$^{-2}$). This indicates that the electric field is well screened by multilayer graphene under such conditions, since the DOS of graphene is finite around the intrinsic Fermi level. The screening in the MoS$_2$ becomes more important only when the Fermi level reach the band edges, that is, when the DOS increases greatly. To verify such statement, we reconstructed the band diagram of 3L MoS$_2$ /1L Graphene system under electric fields of $-2$ V nm$^{-1}$ (Fig. 6e) and $+2$ V nm$^{-1}$ (Fig. 6f). Under $-2$ V nm$^{-1}$ electric field, the Fermi level reaches the CB out the outermost MoS$_2$ layer, while under $+2$ V nm$^{-1}$ electric field, the Fermi level remains within the band gap of MoS$_2$ and has very little shift from the Dirac point of graphene, in good accordance with the ab initio calculations showed in Fig. 5d–f. Charge accumulation occurs mostly on graphene and the outmost MoS$_2$ layer, due to the larger DOS of both layers.

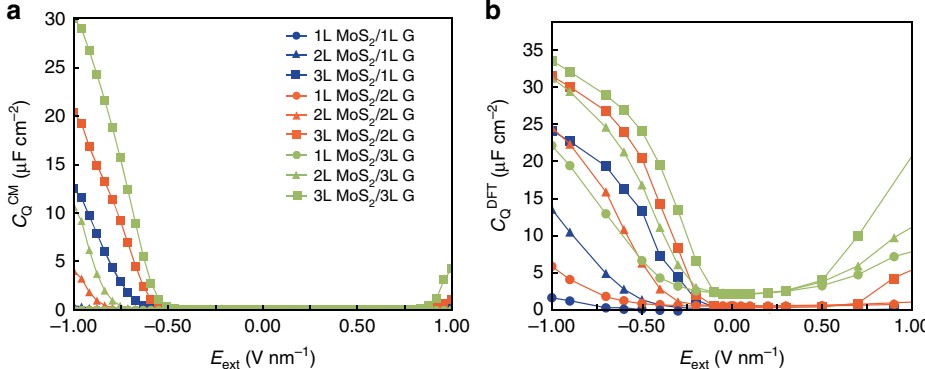

**Fig. 7** Quantum capacitance $C_Q$ for different MoS$_2$/Graphene heterostructures. **a–b** Quantum capacitances calculated from the classical model $C_Q^{CM}$($\mu$F cm$^{-2}$) and from the DOS profile of ab initio calculations $C_Q^{DFT}$($\mu$F cm$^{-2}$) as a function of the external field E$_{ext}$ (V nm$^{-1}$) for various MoS$_2$/Graphene vdWHs, respectively. The quantum capacitance increase more under negative electric field than positive field. Both classical and DFT calculations show similar trends of relationship between layer numbers and quantum capacitance: the layer numbers of MoS$_2$ contributes more to the total quantum capacitance than graphene under strong electric field, consistent with the findings of layer-dependent dipole moment and charge transfer

**Unifying classical and quantum approaches**. The classical electrostatic model shows sound agreement with the ab initio calculations in predicting the asymmetric screening behavior of the MoS$_2$/Graphene vdWH, as a result of the different energy levels and DOS of both materials. Inspired by the equation of EFM response (Eq. 2), that the local capacitance of the vdWHs can be varied under different electric field (see experimental section), here we further propose that such asymmetry can be described by the quantum capacitance, within both theoretical frameworks. The vdWH can be considered as a capacitor, characterized by the quantum capacitance ($C_Q$), which is a function of the DOS at the Fermi level of the whole vdWH: $C_Q = \text{DOS}(E_{Fermi})e^2$. The total DOS of the vdWH can be obtained quantum mechanically at high accuracy. On the other hand, the apparent quantum capacitance in the classical model can be calculated by the differentiating the charge density by the potential drop across the vacuum level $\Delta E_{vac}$ (equivalent to the total difference of work functions $\sum \Delta\phi_i$): $C_Q = \partial Q/\partial \Delta E_{Vac}$. We compare the quantum capacitances calculated by the classical model ($C_Q^{CM}$) and DFT calculations ($C_Q^{DFT}$) as functions of E$_{ext}$ in various MoS$_2$/Graphene systems in Fig. 7a, b, respectively. Interesting, the quantum capacitances calculated by both methods show very similar behavior under external electric field, with the maximum quantum capacitance reaching ~30–33 $\mu$F cm$^{-2}$ in 3L MoS$_2$ /3L Graphene configuration, under an electric field of −1 V nm$^{-1}$. We find the layer dependency of quantum capacitance is very similar to that of the dipole moment and charge transfer: the layer number of MoS$_2$ is dominating the magnitude of the total quantum capacitance under strong electric field. This is reasonable due to the higher quantum capacitance of MoS$_2$ than graphene when Fermi level shifts to the band edges[33]. Note that the DFT calculations predict a non-zero quantum capacitance of vdWHs with 2L and 3L graphene even without external electric field, as a result of the interlayer coupling, which is not included in the classical model. Despite the minute difference between both theoretical frameworks, it is clear that the quantum capacitance of the MoS$_2$/Graphene vdWH, as a combination of the energy levels and DOS, can describe the asymmetric screening behavior with good precision. Our multiscale theoretical framework is thus readily applicable for a variety of vdWHs beyond graphene/MoS$_2$, by utilizing the electronic "genome", in particular, the quantum capacitances of individual 2D layers. A full picture of electric screening of 2D vdWHs can be built benefited from the framework proposed in this work, extending the "tip of the iceberg" of the electrostatic nature of two-layer 2D vdWHs revealed by recent studies[29,38–40].

## Discussion

In summary, our findings reveal fundamental knowledge of the screening properties of van der Waals heterostructures using widely used two-dimensional materials, such as graphene and MoS$_2$. Graphene/MoS$_2$ constitutes an archetypal of vdW heterostructure with exciting possibilities for electronic devices based on atomically thin films. We have shown an asymmetric electric field response on the screening properties of MoS$_2$/Graphene stackings via high-resolution EFM spectroscopy and a multiscale theoretical analysis that involve quantum mechanical ab initio density functional theory, including vdW dispersion forces, and a classical electrostatic approach considering the quantum capacitance. Our ab initio calculations are further unified in a quantum capacitance-based model, showing that the difference between the energy levels and band structures between graphene and MoS$_2$ is account for the asymmetric screening behavior. After the transfer of MoS$_2$ on graphene, the screening of either isolated graphene or MoS$_2$ changes accordingly to the sign of the electric bias utilized becoming polarity-dependent. The EFM phase spectrum shows an asymmetry with the tip voltage, as the number of MoS$_2$ layers increases relative to that of the graphene. Electric fields towards MoS$_2$ tend to be better screened than those directed to graphene as an asymmetrical polarization associated with charge transfer at the MoS$_2$/Graphene interface, generate response fields that opposed to external bias. Such charge rearrangement also polarized the interface inducing the appearance of dipole moments and consequently giving a directional character to the underlying electronic structure. In particular, external fields in such vdW heterostructures can select which electronic states can be used to screen the gate bias, which clearly give an external control on the screening properties according to the stacking order and thickness. Our computational-experimental framework paves the way to understand and engineer the electronic and dielectric properties of a broad class of 2D materials assembled in heterojunctions for different technological applications, such as optoelectronics and plasmonics.

## Methods

**Experimental**. The graphene and MoS$_2$ nanosheets were mechanically exfoliated by Scotch tape. Highly oriented pyrolytic graphite (HOPG) (Momentive, US) and synthetic MoS$_2$ crystals (2D Semiconductors, USA) were used as received. Si wafers with 90 and 270 nm thick thermal SiO$_2$ were used for graphene and MoS$_2$,

respectively. An Olympus optical microscopy (BX51) equipped with a DP71 camera was used to search atomically thin nanosheets, and a Cypher AFM (Asylum Research, US) was employed for topography measurements. The Raman spectra were collected by a Renishaw Raman microscope using 514.5 nm (for $MoS_2$) and 633 nm (for graphene) lasers and an objective lens of ×100(a numerical aperture of 0.9). For the fabrication of the heterostructure, the identified $MoS_2$ nanosheets were firstly coated by thin layer of PMMA, then peeled off from the $SiO_2/Si$ via etching by NaOH, stacked on graphene on $SiO_2/Si$ under the optical microscope. The $MoS_2$/Graphene structure was transferred to Au (100 nm) coated $SiO_2/Si$ substrate following a similar procedure. The Au coating was produced by a Leica ACE600 sputter with a crystal balance monitoring the coating thickness in real time. The EFM measurements were conducted on the Cypher AFM. The EFM phase data were collected by a Pt/Ti-coated cantilever with a spring constant of ~2 N/m (ElectricLever, Asylum Research, USA) dwelling above the heterostructure at a sampling rate of 2 kHz, while a voltage sweeping linearly from +9 to −9 V was applied on the cantilever over a period of 20 s.

**Ab initio quantum calculations**. Calculations were based on ab initio density functional theory using the SIESTA[41] and the VASP codes[42,43]. Projected aug-mented wave method (PAW)[44,45] for the latter, and norm-conserving (NC) Troullier-Martins pseudopotentials[46] for the former, have been used in the description of the bonding environment for Mo, S and C. The shape of the numerical atomic orbitals (NAOs) was automatically determined by the algorithms described in ref. [41]. The generalized gradient approximation[47] along with the DRSLL[48] functional was used in both methods, together with a double-zeta polarized basis set in Siesta, and a well-converged plane-wave cutoff of 500 eV in VASP. We have explicitly checked the effect of different modifications on the exchange part of the vdW density functional on the interlayer distance between graphene and $MoS_2$ as shown in Supplementary Table 1. Minor differences were found between DRSLL and other vdW functionals, as the interlayer distance changes by around 2%. This resulted in negligible variations on the charge density (see Supplementary Note 2 and Supplementary Figure 3). The cutoff radii of the different orbitals in SIESTA were obtained using an energy shift of 50 meV, which proved to be sufficiently accurate to describe the geometries and the energetics. Atoms were allowed to relax under the conjugate-gradient algorithm until the forces acting on the atoms were less than $1 \times 10^{-8}$ eV Å$^{-1}$. The self-consistent field (SCF) convergence was also set to $1.0 \times 10^{-8}$ eV. To model the system studied in the experiments, we created large supercells containing up to 394 atoms to simulate the interface between different number of graphene and $MoS_2$ layers. We have optimized the supercell for the $MoS_2$/Graphene interface using a $5 \times 5$ graphene cell on a $4 \times 4$ $MoS_2$ cell, where the mismatch between different lattice constants is smaller than ~2.0% (i.e. the systems are commensurate). We have kept the lattice constant of the $MoS_2$ at equilibrium, and stretched the one for graphene by that amount. Negligible variations of the graphene electronic properties are observed with the preservation of the Dirac cone for all systems. To avoid any interactions between supercells in the non-periodic direction, a 20 Å vacuum space was used in all calculations. In addition to this, a cutoff energy of 120 Ry was used to resolve the real-space grid used to calculate the Hartree and exchange correlation contribution to the total energy in SIESTA. The Brillouin zone was sampled with a $9 \times 9 \times 1$ grid under the Monkhorst-Pack scheme[49] to perform relaxations with and without van der Waals interactions. Energetics and electronic band structure were calculated using a converged $44 \times 44 \times 1$ **k**-sampling for the unit cell of Graphene/$MoS_2$. In addition to this we used a Fermi–Dirac distribution with an electronic temperature of $k_B T = 20$ meV to resolve the electronic structure.

The electric field $E_{ext}=E_z$ across the vdW heterostructures is simulated using a spatially periodic sawtooh-like potential $V(r) = e\mathbf{E} \cdot \mathbf{r}$ perpendicular to the $MoS_2$/Graphene heterostructures. Such potential is convenient to analyze the response of finite systems (e.g., slabs) to electric fields[22,50–55], while problematic for extended systems (e.g., bulk). The magnitudes of the spatially varying electrostatic potential $<V(z)> = \frac{1}{A}\int_A V(x,y,z)\mathrm{d}x\mathrm{d}y$ and charge density $<\rho(z)> = \frac{1}{A}\int_A \rho(x,y,z)\mathrm{d}x\mathrm{d}y$ across the supercell are determined via a convolution with a filter function to eliminate undesired oscillations and conserve the main features important in the analysis. The variations of both quantities, $\Delta<\rho(z)> = <\rho(z)>_{E\neq0} - <\rho(z)>_{E=0}$ and $\Delta<V(z)> = <V(z)>_{E\neq0} - <V(z)>_{E=0}$ are determined relative to zero field. The polarization $P(r)$ is calculated by the integration of $\Delta<\rho(z)>$ through $\nabla \cdot \mathbf{P} = -\Delta\rho(r)$. $P_{slab}$ is defined through $P_{slab} = \chi E_{slab}$ where $E_{slab} = \frac{E_{ext}}{(1+4\pi\chi(1-l/c))}$, with $l$ the thickness of the vdW heterostructure, and $c$ the height of the supercell[23,24].

**Classical electrostatic model**. In the classical model, the graphene and $MoS_2$ layers are treated as individual layers, with the the band structures (band gap, DOS and intrinsic work functions) considered as invariable to external electric field (i.e. the effect interlayer coupling on band structure is neglected). We further assume that the interlayer distance $d_i$ between the i-1 and i-th layers is fixed, and taken as the interlayer distance in DFT calculations under zero field. The interlayer dielectric constant $\varepsilon_i$ between the i-1 and i-th layer is considered as uniform. For simplicity, we consider that $\varepsilon_i$ is independent of the external field, while the current model can be easily adapted for field-dependent dielectric constant in multilayer 2D materials using ab initio calculation results[23,24]. We consider the interlayer electric field $E_i$ to be uniform. The charge density $Q_i$ and work function $\phi_i$ of each

layer can thus be solved through the following conservation equations in a self-consistent way:

The charge neutrality of the vdWH:

$$\sum_i Q_i = 0 \qquad (4)$$

Charge balance of the $i$-th layer by Gauss law:

$$\varepsilon_i E_i + Q_i - \varepsilon_{i+1}E_{i+1} = 0 \qquad (5)$$

$Q_i$ as a function of the work function $\phi_i$ of $i$-th layer:

$$Q_i(\phi_i) = \int_{-\infty}^{\infty} \text{DOS}(E')\big[f\big(E' - e\phi_i\big)\big) - f\big(E' - e\phi_{i0}\big)\big]\mathrm{d}E', \qquad (6)$$

where $f(E)$ is the Fermi–Dirac distribution function, and $\phi_{i0}$ is the intrinsic work function of the $i$-th layer.

The potential drop between 2 adjacent layers:

$$\Delta_i = \phi_{i+1} - \phi_i = E_i d_i. \qquad (7)$$

Note that for individual layers, $C_{Q,i} = \text{DOS}_i e^2$. We simplify the quantum capacitance of graphene using a linear model: $C_{Graphene,i} = 26.1\ \mu\text{F cm}^{-2}\ \text{V}^{-1}\ \Delta\phi_i$, while the quantum capacitance of $MoS_2$ is a step function, where no density of states exist within the band gap, while $C_{MoS_2,n} = 48\ \mu\text{F cm}^{-2}$ and $C_{MoS_2,p} = 180\ \mu\text{F cm}^{-2}$ for VB and CB, respectively. The intrinsic work function of graphene and $MoS_2$ are set at 4.6 and 4.5 V, respectively. The energy levels of CB and VB of $MoS_2$ are taken as −4.0 and −5.8 eV, respectively.

**Data availability**. The data that support the findings of this study are available from the corresponding author upon reasonable request.

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

## Acknowledgements

We acknowledge valuable discussions with Fritz B. Prinz. C.J.S and T.T. are grateful for financial support from ETH startup funding. L.H.L. thanks the financial support from Australian Research Council (ARC) via Discovery Early Career Researcher Award (DE160100796). E.J.G.S. acknowledges the use of computational resources from the UK national high performance computing service (ARCHER) for which access was obtained via the UKCP consortium (EPSRC grant ref EP/P022626/1); the UK Materials and Molecular Modeling Hub for access to THOMAS supercluster, which is partially funded by EPSRC (EP/P020194/1). The Queen's Fellow Award through the grant number M8407MPH, the Enabling Fund (A5047TSL), and the Department for the Economy (USI 097) are also acknowledged.

## Author contributions

E.J.G.S. and L.H.L. designed the research. L.H.L. fabricated the samples, perform characterization, and EFM measurements. Q.C. performed Raman characterization. T.T. and C.J.S. developed the capacitor model. E.J.G.S. developed the first-principles calculations and performed analysis. L.H.L., T.T. and E.J.G.S. co-wrote the manuscript with inputs from all authors. All authors contributed to this work, read the manuscript, discussed the results, and all agree to the contents of the manuscript.

## Additional information

**Competing interests:** The authors declare no competing interests.

