## [Peer Review File · Nature Communications]

Reviewers' comments:

Reviewer #1 (Remarks to the Author):

In this well-written paper, the authors report on observation of an anisotropic electric response in the MoS₂/graphene van der Waals (vdW) heterostructures in response to the direction of the external electric field. The experiments seem carefully executed and are accompanied with theoretical simulations and models. The finding, as the authors stated, reveal fundamental knowledge of the screening properties of vdW heterostructures and could lead to novel device applications such as optoelectronic and plasmonic devices. The presentation is moreover of a very high quality. I see no objection to the publication of the work in its present form in Nature Communications. However, I do caution the Editor that, as an experimentalist, I might have missed defects in the theoretical simulation/modelling.

Reviewer #2 (Remarks to the Author):

The authors present an interesting study of the electrical field screening of van der Waals heterostructures, combining advanced experimental probes with first-principle theory and an easy-to-grasp classical-electrostatic approach.

The paper presents a convincing demonstration of designable asymmetric electrical field screening of van der Waals heterostructures. This could be of much utility in novel electronic devices based on van der Waals heterostructures.

I believe the paper could trigger several studies of electrical field screening of vdW heterostructures and I am therefore inclined to recommend the paper for publications.

However, I have two concerns:

1) While both indicating asymmetric electrical field screening, the link between theory and experiment is quite weak.

The paper would be more convincing if the authors made a simple quantitative model (damped oscillator model) showing how the computed asymmetric response gives rise to the variation in the EFM phase. A qualitative agreement with some reasonably chosen parameters would be sufficient to make a compelling case.

2)

The choice of xc-functional is unclear and may be inappropriate.

DRSLL is rarely used to denote the van der Waals density functional of Dion et. al. vdW-DF would be more standard.

Note also that vdW-DF is a total xc-energy functional, it is not vdW-"corrected" GGA.

Beyond nomenclature, the standard partner for the non-local correlation of vdW-DF is a revPBE exchange functional; however, this choice significantly overestimate inter-layer separations. The text might be interpreted as if the authors used the PBE exchange functional in combination with vdW-DF non-local correlation. However, this is not a recommended choice and should be avoided in studies like this one.

It would be more appropriate to use for instance vdW-DF-cx or rev-vdW-DF2.

For more details on vdW-DF, consult ROPP 78, 066501, 2015.

The choice of functional is crucial for the specific quantitative predictions as inter-layer separations could have great bearing on the screening properties.

Minor comments:

Why not use the word asymmetric throughout, and not use anisotropic. The reader might believe it refers to anisotropic susceptibility, (i.e. dipole forming in the xy plane), but that is not the topic of this paper.

Some abbreviations are undefined, in particular in the experimental section. Maybe also avoid using G as abbreviations both in Raman spectroscopy and for describing the structure.

Yours sincerely,

Dr. Kristian Berland.

Response to Reviewers

NCOMMS-17-33374A

We thank the reviewers for their detailed attention to our work and their supportive and insightful comments. We have considered each of the comments carefully, and provide point-by-point responses to them below in blue text. Relevant changes to the manuscript are also written in green text.

Reviewer: #1

In this well-written paper, the authors report on observation of an anisotropic electric response in the MoS₂/graphene van der Waals (vdW) heterostructures in response to the direction of the external electric field. The experiments seem carefully executed and are accompanied with theoretical simulations and models. The finding, as the authors stated, reveal fundamental knowledge of the screening properties of vdW heterostructures and could lead to novel device applications such as optoelectronic and plasmonic devices. The presentation is moreover of a very high quality. I see no objection to the publication of the work in its present form in Nature Communications. However, I do caution the Editor that, as an experimentalist, I might have missed defects in the theoretical simulation/modelling.

Reviewer: #2

The authors present an interesting study of the electrical field screening of van der Waals heterostructures, combining advanced experimental probes with first-principle theory and an easy-to-grasp classical-electrostatic approach. The paper presents a convincing demonstration of designable asymmetric electrical field screening of van der Waals heterostructures. This could be of much utility in novel electronic devices based on van der Waals heterostructures.

I believe the paper could trigger several studies of electrical field screening of vdW heterostructures and I am therefore inclined to recommend the paper for publications.

However, I have two concerns:

1) While both indicating asymmetric electrical field screening, the link between theory and experiment is quite weak.

The paper would be more convincing if the authors made a simple quantitative model (damped oscillator model) showing how the computed asymmetric response gives rise to the variation in the EFM phase. A qualitative agreement with some reasonably chosen parameters would be sufficient to make a compelling case.

Response

We thank the reviewer for his/her comments. First and foremost, it should be emphasized that the quantification of any EFM results is very complicated and challenging. This is mainly due to the unknown capacitance term determined by the tip and sample geometry and the long-range nature of electrostatic force. The precise calculation of tip-sample capacitance C is only achievable by numerical methods, and several calibrated parameters *ad hoc* (see Gomila *et al.* Nanotechnology 25, 255702 (2014)). Such analysis is beyond the scope of this project. However, we tried to establish a qualitative agreement between the simulation and EFM experiment using very rough approximations, as requested by the reviewer.

We have included a new set of discussions and data in the *Supporting Information* (1. Qualitative analysis on the EFM results) where we use reasonable parameters to strengthen the link between theory and experiments. We could estimate the behavior of the electric susceptibility as a function of the bias for one of our systems, 3L MoS₂/4L Graphene. It can be seen in Figure S2 that there is a systematic increment of the electric susceptibility as the bias points towards MoS₂, which follows the behavior observed in the simulations.

We added the following sentences to the manuscript:

Page 7, line 152: “We tried to qualitatively analyze the EFM results to estimate the change of electric susceptibility (χ) of the 3L MoS₂/4L Graphene heterostructure (see *Supporting Information*). The behavior displays an increment of χ with the bias pointing to the MoS₂ surface.”

Page 9, line 178: “...(Fig. 2(f) and Fig. S2 in the Supporting Information) ...”

2) The choice of xc-functional is unclear and may be inappropriate.

DRSLL is rarely used to denote the van der Waals density functional of Dion et. al. vdW-DF would be more standard.

Note also that vdW-DF is a total xc-energy functional, it is not vdW-"corrected" GGA.

Beyond nomenclature, the standard partner for the non-local correlation of vdW-DF is a revPBE exchange functional; however, this choice significantly overestimate inter-layer separations. The text might be interpreted as if the authors used the PBE exchange functional in combination with vdW-DF non-local correlation. However, this is not a recommended choice and should be avoided in studies like this one.

It would be more appropriate to use for instance vdW-DF-cx or rev-vdW-DF2.

For more details on vdW-DF, consult ROPP 78, 066501, 2015.

The choice of functional is crucial for the specific quantitative predictions as inter-layer separations could have great bearing on the screening properties.

Response

We have explicitly checked the effect of different van der Waals (vdW) density functional (DF) on the magnitude of the interlayer distance between graphene and MoS₂ layers (d_{G-MoS_2}). We have used four different DF's, including the one suggested by the reviewer, e.g. BH. We found minor modifications between the vdW-DF used in our manuscript (DRSLL) and others with different changes on the exchange part. The magnitudes of d_{G-MoS_2} differ by around 2%, which also compare well with recent values reported in the literature^{1,2,3}. We also calculated the charge density using these new vdW functionals and negligible variations were observed. This fully justifies our choice of DRSLL in our simulations.

We have included these results in the *Supporting Information* (2. Interlayer distance between graphene and MoS₂ sheets), in Table S1, Figure S4, together with discussions, comparison with literature and new references. We have included in page 28, line 502, the following sentences referring to this analysis:

“We have explicitly checked the effect of different modifications on the exchange part of the vdW density functional on the interlayer distance between graphene and MoS₂ as shown in Table S1 in the *Supporting Information*. Minor

differences were found between DRSSL and other vdW functionals, as the interlayer distance changes by around 2%. This resulted in negligible variations on the charge density (Figure S4).”

We have also corrected the text to show that we used a total energy functional as suggested by the reviewer.

In page 4, line 70: “...we employed quantum mechanical *ab initio* simulations based on density functional theory (DFT) using a total energy vdW functional to resolve ...”

Page 8, line 155: “...can be explained at two theoretical levels, both by *ab initio* simulations with vdW- functionals, ...”

1. Y. Ma, Y. Dai, M. Guo, C. Niu and B. Huang. Graphene adhesion on MoS2 monolayer: An ab initio study. *Nanoscale* 3, 3883 (2011).
2. W. Hu, T. Wang, R. Zhang and J. Yang. Effects of interlayer coupling and electric fields on the electronic structures of graphene and MoS2 heterobilayers. *J. Mat. Chem. C* 4, 1776 (2016).
3. X. Liu and Z. Li. Electric Field and Strain Effect on Graphene-MoS2 Hybrid Structure: Ab Initio Calculations. *J. Phys. Chem. Lett.* 6, 3269 (2015).

3) Minor comments:

Why not use the word asymmetric throughout, and not use anisotropic. The reader might believe it refers to anisotropic susceptibility, (i.e. dipole forming in the xy plane), but that is not the topic of this paper.

Some abbreviations are undefined, in particular in the experimental section. Maybe also avoid using G as abbreviations both in Raman spectroscopy and for describing the structure.

Response

We have reviewed the manuscript thoroughly using the word “asymmetric” instead of “anisotropic” throughout the text.

We have also defined unclear abbreviations in the text.

Page 4, line 87: “achieved by two rounds of polymethyl-methacrylate (PMMA) transfer...”

Page 5, line 99: “The corresponding atomic force microscopy (AFM) image of the heterostructure on Au is displayed ...”

Page 6, line 117: “ V_{CPD} is the contact potential difference (CPD) due to the mismatch ...”

We have modified throughout the text the abbreviation of “graphene” by “G”, and now use “graphene” instead for describing the structures.

REVIEWERS' COMMENTS:

Reviewer #2 (Remarks to the Author):

The authors have addressed both concerns in a satisfactory manner. I recommend it for publication.